# Speaker Anonymization: Disentangling Speaker Features from Pre-Trained Speech Embeddings for Voice Conversion

**Marco Matassoni [1], Seraphina Fong [2] and Alessio Brutti [1,*]**

[1]  Augmented Intelligence Center, Fondazione Bruno Kessler, 38100 Trento, Italy; matasso@fbk.eu
[2]  Department of Psychology and Cognitive Science, University of Trento, 38068 Rovereto, Italy;
    meiyueseraphina.fong@unitn.it
*  Correspondence: brutti@fbk.eu

**Abstract:** Speech is a crucial source of personal information, and the risk of attackers using such information increases day by day. Speaker privacy protection is crucial, and various approaches have been proposed to hide the speaker's identity. One approach is voice anonymization, which aims to safeguard speaker identity while maintaining speech content through techniques such as voice conversion or spectral feature alteration. The significance of voice anonymization has grown due to the necessity to protect personal information in applications such as voice assistants, authentication, and customer support. Building upon the S3PRL-VC toolkit and on pre-trained speech and speaker representation models, this paper introduces a feature disentanglement approach to improve the de-identification performance of the state-of-the-art anonymization approaches based on voice conversion. The proposed approach achieves state-of-the-art speaker de-identification and causes minimal impact on the intelligibility of the signal after conversion.

**Keywords:** privacy protection; anonymization; voice conversion; voice cloning

## 1. Introduction

An enormous amount of voice data is being produced by the widespread adoption of voice technologies in our daily lives. Besides the use of speech biometrics for authentication [1], voice instructions to virtual assistants, and voice communications across multiple industries, social media is also causing an exponential increase in the amount of speech data available online [2]. This trend raises some serious issues, most notably those pertaining to security and privacy [3]. Voice recordings have the potential to disclose personal information such as age, gender, ethnic origin, and religious beliefs [4]. Modern voice biometrics have the capability to derive a variety of the aforementioned private and sensitive data from a speech signal. Given these concerns, and also as voice assistants and voice recognition technologies become more ubiquitous, protecting user identities within voice data is becoming increasingly vital.

This has led to an increased interest in privacy preservation solutions for speech technology that, following the General Data Protection Regulation (GDPR) introduced by the European Union [5], aim to protect personal speech data. One popular approach to mitigate these issues is to develop privacy by design solutions, which perform the data processing on the edge devices and are close to where the data are generated. This solution avoids transferring or storing sensitive information on cloud infrastructures. However, the limited computational resources of edge nodes typically require the compression of AI models in order to reduce their memory and computation requirements [6–8]. In addition, this approach often requires the design of tailored solutions, which may not always be possible.

Alternative approaches are based on encryption [9,10]. Similarly, implementing jamming mechanisms on speech signals may obfuscate the user's identity [11,12]. There are

two main limitations of encryption: (1) the computational cost and (2) that the cloud entity processing the data, in any case, has access to the raw signals.

Recently, techniques for voice conversion (VC), which enable the alteration of one's voice into another, have become a viable means of accomplishing privacy preservation via speech anonymization [13–16]. Speaker anonymization is a user-centric voice privacy solution that aims to remove speaker information from a speech utterance while leaving the other acoustic attributes unaltered. In this way, it is possible to conceal a speaker's identity without compromising intelligibility and naturalness. Voice anonymization offers a transparent solution for the following speech processing strategies and does not have to be tailored or adapted to a specific privacy-preservation method. Figure 1 graphically shows these two strategies: privacy by design and voice anonymization.

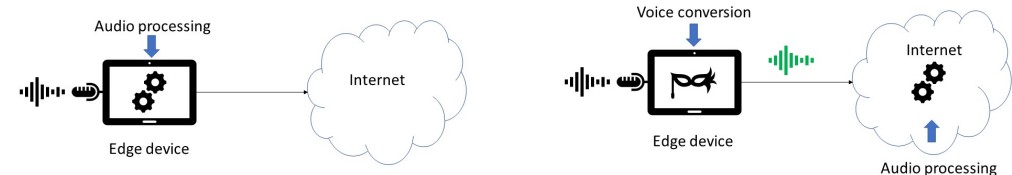

(**a**) Privacy-by-design approach.          (**b**) Audio anonymization by voice conversion.

**Figure 1.** Preserving privacy strategies in cloud-based speech applications (from [17]).

The VoicePrivacy Challenge [18] in 2020 and 2022 accelerated advancements in speaker anonymization techniques. Most approaches in the VoicePrivacy Challenge can be categorized into signal-processing-based voice transformation and voice conversion based on speaker embeddings [14,19–25]. Signal anonymization based on processing does not require training data and directly alters speech characteristics like the time scale, the pitch, or the spectral envelope. As the range of physical shifts in voice signals is limited, attackers may be able to recover the original speech after a certain number of tries. Therefore, many modern anonymization systems are based on the process of separating the speaker-related representation from the raw speech signal using neural paradigms. These methods reduce speaker-specific characteristics by averaging or changing the speaker embeddings, which are often achieved using representations retrieved from a pre-trained automated speaker verification model. The final result is the disentanglement between the speech content and the speaker information [26–28].

In spite of the success of these latter methods, there are still a great deal of areas that might be improved, such as making anonymized speech more distinctive and handling more potent assault scenarios [13,29–32]. Notably, the forthcoming VoicePrivacy Challenge will focus on developing attack models against speaker anonymization systems (https://www.voiceprivacychallenge.org/ (accessed on 1 March 2024)). A further limitation of the current approaches is the trade-off between the amount of anonymization and the quality of the processed speech. As a matter of fact, if the speech embeddings carry a great deal of acoustic information to allow for high-quality speech synthesis, they inevitably include information about the speaker. Refs. [33–35] discussed some of the aspects related to the residual speaker information preserved in anonymization mechanisms.

As a follow up to our prior effort, as presented in [17], this work proposes a novel strategy for anonymization via voice conversion, which, instead of manipulating the x-vectors, leverages the approach of ContentVec [36] to obtain speaker-independent speech representations and starts from pre-trained models within the S3PRL toolkit [37]. The proposed strategy is evaluated on a public dataset and compared against a variety of neural and signal-processing-based voice conversion methods.

*Significance of Speaker Anonymization*

There are many contexts where speaker anonymization plays a crucial role:

- **Privacy Protection and Ethical Data Use** The preservation of personal privacy is critical in a time when personal data are increasingly being stored digitally. If voice recordings are not sufficiently anonymized, they can be used for malicious activities such as identity theft and spying. Voice anonymization techniques allow voice data to be used while hiding the speaker's identity.
- **Legal Compliance** Organizations handling voice data are required to abide by the increasingly strict data protection regulations being introduced by governments and regulatory bodies. In order to guarantee that sensitive voice data are handled in a way that complies with legal requirements, voice anonymization can be a crucial part of compliance strategies.
- **Enhanced Security** Malicious actors may use exposed voice data to commit fraud or identity theft. The risks of data breaches and unauthorized access are reduced by anonymizing voice recordings.
- **Versatile Applications** In addition to protecting one's privacy, voice anonymization makes it possible to use voice data for a variety of purposes without sacrificing the data's security or confidentiality. This covers everything from enhancing call center and voice assistant functionality to supporting law enforcement organizations in protecting the privacy of their witnesses.

In summary, removing voice prints from recorded audio data is crucial from a variety of technological and ethical points of view. Nevertheless, voice anonymization presents important ethical challenges. For instance, one key concern is the potential misuse of these techniques, such as in deepfake scenarios where malicious actors may use voice conversion to impersonate others. Striking a balance between the legitimate use of voice anonymization and preventing unethical use is a complex ethical dilemma. Furthermore, the issue of informed consent is paramount. When voice data are collected, individuals should be informed about how it will be used, including whether or not speaker anonymization will be applied.

## 2. State-of-the-Art Techniques

As mentioned above, privacy preservation for speech can be achieved through several techniques: obfuscation, encryption, distributed learning, or anonymization. Obfuscation suppresses or modifies speech signals [38], while encryption supports computation in encrypted data but increases computational complexity. Decentralized learning trains models from distributed data without directly accessing it [39]. However, information about original data may be leaked [40], unless further security mechanisms are applied [41,42].

The basic techniques for voice anonymization rely on signal-processing methods [43], which modify instantaneous speech characteristics such as pitch [44], spectral envelope [45], and time scaling [46]. For example, [45] utilized McAdams coefficients to randomize the formants of speech signals. A similar but more articulated approach is presented in [38], which explored vocal anonymization in urban sound recordings, thereby aiming to obscure speech content and de-identify the speaker while preserving the rest of the acoustic scene. The approach, inspired by face blurring in images, separates the audio signal into voice and background components, and it selectively distorts the voice signal before mixing it back with the background.

Current anonymization approaches mainly focus on learning specialized latent representations, which decompose speech into content, speaker identity, and prosody. The speaker identity is a statistical time-invariant representation throughout an utterance, while content and prosodic information vary over time. The speaker's identity representation carries most of the private information, so generated speech using original content, prosodic, and anonymized speaker representations can suppress the original identity information while maintaining intelligibility and naturalness.

A representative technique is discussed in [19], where a bottle-neck feature extractor is combined with a sequence-to-sequence synthesis module that maps the spectral features from the bottle-neck features under an additional constraint that governs the resulting

speaker representations. A general framework in this direction involves the following: (i) fine-grained disentangled representation extraction from original speech, (ii) speaker representation anonymization, and (iii) anonymized speech synthesis. The selection-based speaker anonymizer is the most significant performance bottle neck for the current mainstream approach, as it depends on the distribution of the external pool of speakers and how pseudo-speakers are selected. This pipeline has been widely used in recent works [47,48], but it does not prevent speaker-related information and linguistic features from leaking into the anonymized waveform during vocoding. In general, a trade-off has to be found between the quality of the reconstructed speech and the degree of leaked speaker information. Ref. [49] proposed a new synthesis pipeline to avoid the leakage of information between different speech components. Ref. [50] proposed the use of very high-level semantic tokens and acoustic tokens, which can be decoded to resynthesize an audio signal. Differential privacy [26] and feature quantization have been explored to sanitize the speaker information from linguistic features. Alternatively, some methods operate at the speaker-embedding level. Ref. [51] proposed an x-vector anonymization method, based on autoencoders and adversarial training, which transforms the original x-vector into a new one with suppressed speaker characteristics. Analogously, the speaker anonymization system in [52] generates distinctive anonymized speaker vectors that can protect privacy under all attack scenarios, and it can successfully adapt to unseen-language speaker anonymization without severe language mismatch.

More recently, several approaches have focused on the use of pre-trained speech representations (e.g., HuBERT, Wav2vec 2.0, and WavLM). For example, Ref. [53] investigated a vector-matching- and latent-transformation-based type of speaker anonymization, which is inspired by a voice conversion model based on the K-nearest neighbors algorithm (kNN-VC) [54]. This method randomly samples multiple speakers and interpolates them to achieve speaker anonymization. Similarly, Ref. [55] presented an anonymization method based on reprogramming large self-supervised learning (SSL) models to effectively hide speaker identity.

In contrast, a completely different approach was presented in [56], where adversarial examples were used to trick a speaker verification system while inducing imperceptible differences to the human auditory system.

Finally, Ref. [57] proposed a two-step procedure, where the anonymization model described in the Voice Privacy Challenge 2022 was complemented with a zero-shot voice conversion block, thereby demonstrating strong privacy properties combined with strong pitch information preservation.

## 3. The Proposed Approach: Disentanglement of Pre-Trained Speech Embeddings

Our approach is built upon the work of [58] (S2VC) for voice conversion, in which the speech content of an input audio signal, expressed as bottle neck features related to an Automatic Speech Recognition (ASR) task, was learned and represented. An x-vector representation of the target speaker was then used to re-synthesize the audio waveforms. Building upon that concept, we improve S2VC, specifically for speaker anonymization, in two directions:

1. Instead of using posteriorgrams features, as proposed in the original architecture, pre-trained speech and speaker representations are employed.
2. We introduce the use of disentanglement techniques, following a similar direction as ContentVec [36], on pre-trained speech representation.

Figure 2 depicts the voice conversion scheme of the present work. For the speech embeddings, WavLM learns universal speech representations from massive unlabeled speech data and effectively adapts them across various speech processing tasks [59]. As for the speaker feature, a strong representation is provided by the TitaNet model proposed by NVIDIA [60]; it features 1D depth-wise separable convolutions with Squeeze-and-Excitation layers, with the global context followed by a channel-attention-based statis-

tics pooling layer (https://catalog.ngc.nvidia.com/orgs/nvidia/teams/nemo/models/titanet_large (accessed on 1 March 2024)).

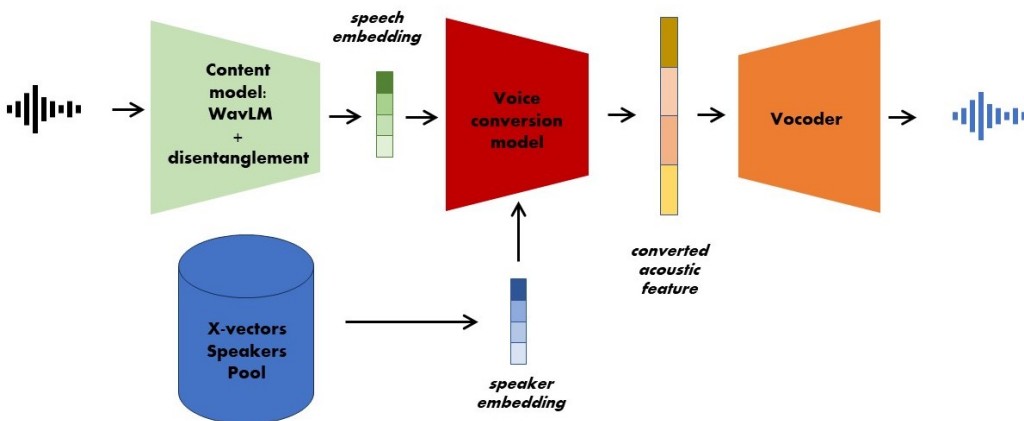

**Figure 2.** Voice conversion scheme: the input signal is encoded with a self-supervised learning (SSL) pre-trained model (i.e., WavLM); the conversion model generates a converted representation using this content representation along with a different speaker embedding; and, finally, the vocoder synthesizes the converted waveform.

Based on our comprehensive analysis, as reported in [17], it was observed that although these speech embeddings produce higher-quality anonymized signals, they still carry residual information about the speaker. Therefore, we introduced the use of disentanglement techniques on the pre-trained speech representation, following a similar direction as ContentVec (https://github.com/auspicious3000/contentvec (accessed on 1 March 2024)) [36], which has been developed to improve ASR performance by focusing all the embedding information on the speech content rather than on the speaker characteristics.

The idea of feature disentanglement is explored in various works [61–64], and it has been aimed at encoding only the specific properties of an incoming speech signal (e.g., for altering prosody and expressivity); in the presented scenario, we investigate the possibility of minimizing the speaker information provided by popular SSL representations. We, therefore, summarize the investigated approach in the anonymization domain. The disentanglement modifies the entire hidden-state representation, thereby learning a suitable transformation that minimizes the differences due to speaker characteristics.

Figure 3 depicts the disentanglement mechanism of the present work. In order to modify only the speaker identity of the utterance, the signals are altered using the modifications described in [36]. The proposed learning mechanism adds an additional cosine similarity loss in the fine-tuning procedure. This forces the ASR model to generate similar embeddings when applied to the original signal and to its altered version. In this way, the model equipped with this additional similarity loss learns to minimize, in the resulting embedding, the information associated to pitch or formants, and—in turn—to speaker characteristics, as illustrated by the results reported in Section 5. As shown in Figure 3, the Connectionist Temporal Classification (CTC) loss on an ASR downstream task (i.e., Librispeech) is also used to preserve the speech content representation capabilities of WavLM while the speaker information is removed.

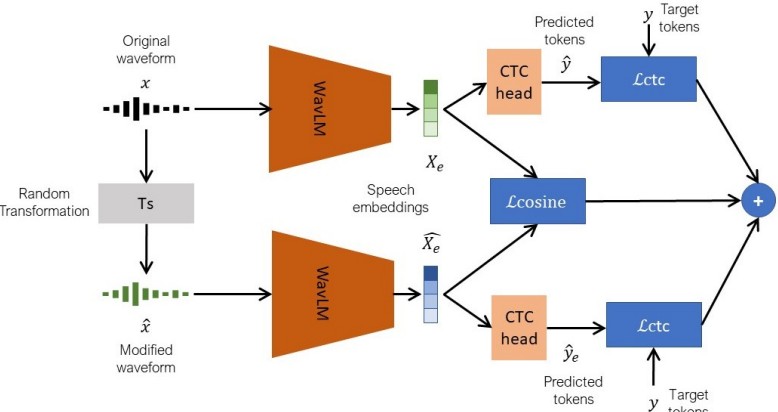

**Figure 3.** Scheme for the disentanglement mechanism: the original signal and a modified version (with pitch and formant shifts) are used as pairs for training the downstream task (e.g., Automatic Speech Recognition). Aside from the original Connectionist Temporal Classification (CTC) loss, an additional loss based on a cosine similarity on the signal pair induces an additional constraint in the resulting representations, which forces the two representations to be similar and hence progressively reduce the information associated with the speaker characteristics.

Let us denote the input waveform as $x$. After applying a set of random signal-based transformations $T_s$, we obtain a modified waveform with the same spoken content and prosody but different speaker characteristics as follows:

$$\hat{x} = T_s(x). \tag{1}$$

Specifically, $T_s(\cdot)$ randomly alters the formant frequencies and pitch, as discussed in [65], as follows:

- the pitch is altered by a random shift in range $[1/1.4 - 1.4]$ and by a random scale in the range $[1/1.5 - 1.5]$;
- similarly, the formants are shift by a random value in the range $[1/1.4 - 1.4]$.

As in the original paper, we used the ParselMouth Python library (https://parselmouth. readthedocs.io/en/stable/ (accessed on 1 March 2024)).

Given the encoder part $Enc(\cdot)$ of the pre-trained WavLM model, we denote the latent representations of the original and modified waveform as $X_e = Enc(x)$ and $\hat{X}_e = Enc(\hat{x})$, respectively. These two representations share the same content but exhibit two different speaker identities. Therefore, the model is fine-tuned using a contrastive loss that penalizes the dissimilarity between $X_e$ and $\hat{X}_e$ as follows:

$$\mathcal{L}_{\text{cosine}}(X_e, \hat{X}_e) = \frac{X_e \hat{X}_e}{\|X_e\| \|\hat{X}_e\|}. \tag{2}$$

The cosine loss above is then combined in the fine-tuning process with the traditional CTC loss, which is computed on both versions on the input waverform as follows:

$$\mathcal{L} = \alpha \mathcal{L}_{\text{cosine}}(X_e, \hat{X}_e) + (1 - \alpha)[\mathcal{L}_{\text{CTC}}(y, \hat{y}) + \mathcal{L}_{\text{CTC}}(y, \hat{y}_e)], \tag{3}$$

where $y$ are the ground-truth tokens, while $\hat{y}$ and $\hat{y}_e$ are the predicted tokens obtained from the original and modified waveforms, respectively.

Once the disentanglement model is trained, the resulting speech representation is used as an upstream model in the S3PRL-VC toolkit [66]. The S3PRL-VC toolkit was proposed to help compare the SSL model performance against the SUPERB-SG [67,68] standard framework. Specifically, it allows us to evaluate the generative capabilities of pre-trained models, as well as the generalizability of the resulting conversion model.

The resulting anonymization task was mainly derived from the setup proposed in [37] for voice conversion. The speech embeddings were computed using the introduced disentanglement mechanism on the WavLM features in the present work. The anonymization model was then trained after the preliminary fine tuning of a pre-trained WavLM model to a recognition task (i.e., with the usual CTC loss), and it was augmented with an illustrated contrastive loss that penalizes the dissimilarities between the signal pairs due to speaker characteristics (see Figure 3).

## 4. Experimental Settings

Three popular public corpora were selected for our experimental analysis: LibriSpeech [69], LibriTTS [70], and the voice cloning toolkit dataset (VCTK) [71]. LibriSpeech contains ≈1000 h of read audiobooks, which were partitioned into ≈960 h for training and ≈20 h for evaluation. LibriTTS is a corpus of ≈585 h of read English speech, specifically designed for TTS and derived from LibriSpeech. The VCTK corpus includes read speech by 110 native English speakers, with each of them providing about 400 sentences. The models were fine-tuned using the LibriTTS *train-clean* sets and the whole VCTK dataset. The LibriSpeech *test-clean* and *test-other* sets (which comprise 4 and 33 speakers, respectively) were used for evaluation. An overview of the dataset partitioning for the training and evaluating of the models is included in Table 1.

**Table 1.** Statistics of the datasets used for training and evaluation. The present work considers LibriSpeech [69], LibriTTS [70], and the voice cloning toolkit dataset (VCTK) [71].

| Corpus | LibriTTS | | LibriSpeech | | VCTK |
|---|---|---|---|---|---|
| Set | Train | Dev-Clean | Test-Clean | Test-Other | Train |
| Speakers | 2311 | 40 | 40 | 33 | 109 |
| Utterances | 354,780 | 5736 | 2620 | 2939 | 44,242 |
| Size | 555 h | 9 h | 5.4 h | 5.3 h | 44 h |

Diverse attack scenarios are discussed in [72], thereby showing the impact of an attacker's knowledge of the achievable protection. Specifically, the strongest attack model ("informed") assumed a complete understanding of the speaker anonymization method by the adversary. However, this type of analysis is beyond the scope of the present work. The current work focuses on the simple ("ignorant") setup, where the attacker is not aware of the anonymization processing.

Differently from [26], our ASR setup is based on a Kaldi-based (fixed) model trained on the original LibriSpeech clean signals; although the resulting ASR may exhibit lower performance, the decision to avoid training on anonymized speech was supported by a more equitable comparison across the analyzed conversion techniques, whereby the methods that exhibited a lower acoustic mismatch with respect to the original audio were favored.

### 4.1. Implementation Details

The disentangled SSL model was trained for three epochs with a learning rate equal to $5 \times 10^{-5}$ and $\alpha = 0.99$, as shown in Equation (3). The conversion model was then trained using the standard procedure described in [66] i.e., 50,000 iterations with a learning rate equal to $1 \times 10^{-4}$. The synthesizer model was the Taco2-AR, which is derived from Tacotron 2 [73].

### 4.2. Metrics

As already proposed in earlier works, the primary objective metrics used were related to privacy and utility [18,74]. As such, for the speaker identification/verification, the Equal Error Rate (EER) is used as the classic metric, which is defined as the error rate corresponding to a certain threshold for which the false alarm rate is equal to the the false miss rate.

For privacy preservation purposes, higher values indicates better anonymization. The EER is measured by employing the NVIDIA speaker recognition baseline using TitaNet [60].

Similarly, for utility, the classic Word Error Rate (WER) was adopted. In this case, lower values are preferred as they would indicate there are lower speech distortions introduced in the generated waveforms. As the present work aims to measure the differences in performance, a Kaldi standard recipe was used as the back-end for ASR (https://kaldi-asr.org/models/m13 (accessed on 1 March 2024)). Decoding was performed with the conventional factorized time delay neural network (TDNN) acoustic model and the associated (*small*) trigram language model [69].

Other measures can be used to evaluate the perceptual quality of the generated speech such as, for example, the Perceptual Evaluation of Speech Quality (PESQ) metric [75] or the Short-Time Objective Intelligibility (STOI) metric [76]. However, these metrics often do not correlate well with ASR performance. Additionally, since the acoustic similarity was evaluated against a reference clean signal, these metrics were found to not be suitable in our anonymization context (i.e., changing the speaker properties inevitably compromises the acoustic similarity).

### 4.3. Voice Conversion Baselines

The first baseline considered was the method based on the **McAdams** coefficient [45], which is a purely signal-processing approach. The power spectrum of a short-term segment of speech is fit with an all-pole model using linear predictive coding analysis. The formant frequencies were determined by the angles of the corresponding complex poles. The speaker anonymization process was, therefore, achieved by varying the pole angles to shift the formants using the McAdams coefficient. Another signal-based technique used as a baseline applies **pitch and formant** changes and resembles the procedure based on the popular Praat toolkit [77]. Given the efficacy of the speech modification approaches used in ContentVec [36], we also consider it as a signal-processing baseline. Three modifications were applied to the signals, i.e., a random shifting of the formants (achieved by scaling by a random factor $\rho_1$), random pitch scaling (by a random factor $\rho_2$), and random frequency shaping using a parametric equalizer [65].

The next baseline considered was the Activation Guidance and Adaptive Instance Normalization-VC method **(AGAIN-VC)** [78], which is an auto-encoder-based model that has a single encoder and a decoder. The application of activation as a bottleneck on content embeddings greatly improves the trade-off between the synthesis quality and the speaker similarity in the converted speech. The related code is publicly available (https://github.com/KimythAnly/AGAIN-VC (accessed on 1 March 2024)). Finally, the original Phonetic Posteriorgram-based VC method **(PPG-VC)** [19] was also considered and included as a baseline. In this case, phonetic posteriors were used as content representations, and the speaker information was therefore effectively obfuscated.

We also considered models based on our approach [17], i.e., ones that encode the speech content through the popular SSL representation, such as **HuBert** [79] or **WavLM** [59]. The last baseline model was based on the aforementioned S3PRL toolkit and a standard WavLM representation (https://huggingface.co/patrickvonplaten/wavlm-libri-clean-100h-base-plus (accessed on 1 March 2024)).

### 4.4. Vocoder

The final vocoder plays a critical role in ensuring a high-quality speech resynthesis in the context of speech anonymization, and this is irrespective of the de-indentification processes implemented beforehand. The present work adopts HiFi-GAN [80]: a state-of-the-art vocoder that consists of one generator and two discriminators (multi-scale and multi-period), which are trained in an adversarial fashion along with two additional losses for improving training stability and model performance. The generator is a fully convolutional neural network. Specifically, it uses a mel-spectrogram as the input, and it upsamples it through transposed convolutions until the length of the output sequence

matches the temporal resolution of the waveform. In this work, the synthesis model was trained using LibriTTS and VCTK together with the speech disentanglement mechanism.

## 5. Experimental Results

By also leveraging the results obtained from our prior investigation in [17], this section reports the performance of our newly proposed disentanglement approach in comparison with the state-of-the-art methods described in Section 4.3. The results were computed on the test-clean partition of LibriSpeech. "Original" in both Tables 2 and 3 refers to the original audio signals without conversion.

Firstly, considering our framework presented in [17], Table 2 presents the performance of the different pre-trained models (HuBert and WavLM) and the following two types of representations: one based on the last hidden layer and another using the posteriorgram produced by the CTC head. As expected, the results confirmed that these representations partially mask speaker's characteristics but still maintain some speaker information. A trade-off exists between the quality of the reconstructed waveform, which is represented by the WER, and the degree of anonymization obtained. Using the output of the CTC head as speech representation leads to very good anonymization, albeit at the cost of a degradation in terms of the speech quality as it only includes information about the speech content. Conversely, using a latent representation inevitably leaks information about the speaker, particularly when larger vectors are used. As per the reminder encouraged from our experimental analysis, we considered WavLM HS-768 as speech embeddings.

**Table 2.** Anonymization performance using Hubert and WavLM as pre-trained models and by considering either the exit of the last hidden layer (HS) or the posteriorgram provided by the CTC head as speech embeddings. The number near each model indicates the size of the speech embeddings. "Original" refers to the original audio signals without conversion.

| Method | EER ↑ | WER ↓ |
|---|---|---|
| Original | 1.2 | 5.29 |
| Hubert CTC-32 | 44.3 | 9.47 |
| WavLM CTC-31 | 44.6 | 9.39 |
| Hubert HS-1024 | 35.2 | 7.38 |
| WavLM HS-768 | 33.5 | 7.50 |

Table 3 reports the performance of our proposed disentanglement approach when we considered the following two different implementations: one that uses WavLM for the speech content representation, while the other is the original S3PRL. The latter implementation was used with and without disentanglement. Firstly, by focusing on the signal-processing methods, we can observe that applying the transformations used in [36] provides very interesting results at a low computational cost. The performance was found to be significantly better than the McAdams, i.e., the other signal-processing method. Traditional voice conversion methods (i.e., AGAIN-VC or PPG-VC) offer very good de-anonymization performance but at the cost of a high degradation of the speech signal. This degraded quality in the generated waveform may be due to a distributional mismatch between the acoustic features predicted by the traditional conversion models and those that the vocoder expects based on its training [24].

Using the S3PRL framework with WavLM HS-768 notably improves the performance with an EER of 37.9% while introducing only a marginal degradation in WER. Finally, the last row in Table 3 shows the considerable benefit provided by the present work's disentanglement approach, as it provided the highest EER (almost 50%) compared to all the other methods. Notably, it significantly increased performance compared to the method without disentanglement (from 37.9% to 49.4% EER) while maintaining a comparable WER (from 6.24 to 6.56). This finding suggests that the disentanglement mechanism is a viable approach to strike a balance between reducing the amount of speaker information present in a signal while also maintaining reasonable intelligibility.

**Table 3.** Results from using various Voice Conversion techniques on the Librispeech *test-clean* set. The considered techniques were two signal-based approaches (McAdams and the alteration of pitch and formants); two state-of-the-art voice conversion techniques (AGAIN-VC and PPG-VC); models based on Self-Supervised Learning (SSL) representations that consider the last internal layer (Hidden Size 1024 or 768) or the CTC head (size 32 or 31) as content encoders; and finally, a WavLM model using S3PRL-VC [66] were trained, with the disentanglement procedure further boosting the performance. "Original" refers to the original audio signals prior to conversion.

| Method | EER ↑ | WER ↓ |
|---|---|---|
| Original | 1.2 | 5.29 |
| McAdams [45] | 5.6 | 11.74 |
| Pitch/Formants shift [36] | 35.2 | 10.27 |
| AGAIN-VC [78] | 44.4 | 44.37 |
| PPG-VC [19] | 46.1 | 12.93 |
| WavLM HS-768 [17] | 33.5 | 7.50 |
| S3PRL [66] using WavLM HS-768 | 37.9 | 6.24 |
| +Disentanglement | **49.4** | 6.56 |

## 6. Conclusions

Voice anonymization, as it is driven by voice conversion techniques, plays a pivotal role in addressing the complex intersection of privacy, security, and ethical concerns associated with the ever-expanding realm of voice data. As the adoption of voice technology continues to grow, the demand for effective voice anonymization solutions will only intensify. The present work has provided an overview of the importance of voice anonymization and the primary methods investigated in possible applications. One important approach is voice conversion, which makes use of self-representation learning to provide a strong framework for addressing the difficulties associated with speaker anonymization. Following a thorough examination of various methods, a promising strategy based on feature disentanglement was created and discussed. The goal of the proposed strategy is to conceal the speaker information contained in speech signals while maintaining content and comprehensibility using a specifically trained speech encoder. The experimental results validate the approach, although a more comprehensive analysis on the quality of the generated signals is suggested. Similarly, a specific study on residual speaker characteristics embedded in the learned representation would permit a fine-grained control of the converted speech when compared to the very recent VoicePAT framework [81] selected for the incoming VoicePrivacy Challenge 2024.

A more complete anonymization speech pipeline may also include a redaction or obfuscation of specific content, such as, for example, spoken named entities [82,83]. However, while the discussed techniques are essentially language-independent, audio de-identification is a more recent field of study that depends on the spoken language being used and requires the integration of multilingual ASR models to cope with general speakers.

The implementation of the disentanglement procedure will be made available in a public repository.

**Author Contributions:** Conceptualization and Methodology, M.M. and A.B.; software, validation, formal investigation, S.F. and M.M.; writing, S.F., M.M., A.B.; funding acquisition, A.B. All authors have read and agreed to the published version of the manuscript.

**Funding:** This work was partially funded by the EU H2020 project MARVEL (project number: 957337).

**Institutional Review Board Statement:** Not applicable.

**Informed Consent Statement:** Not applicable.

**Data Availability Statement:** The original contributions presented in the study are included in the article, further inquiries can be directed to the corresponding authors.

**Conflicts of Interest:** The authors declare no conflict of interest.

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
