# Peer review of "Speaker Anonymization: Disentangling Speaker Features from Pre-Trained Speech Embeddings for Voice Conversion"

_applsci, doi:10.3390/app14093876_

Round 1
Reviewer 1 Report
Comments and Suggestions for Authors
Speech is a crucial source of personal information, and the risk of attackers using such information increases day by day. Speaker privacy protection is crucial, and various approaches have been proposed to hide the speaker’s identity. One approach is voice anonymization, which aims to safeguard speaker identity while maintaining speech content through techniques voice conversion or spectral feature alteration. The significance of voice anonymization has grown due to the necessity to protect personal information in applications such as voice assistants, authentication, and customer support. Building upon the S3PRL-VC toolkit and on pre-trained speech and speaker representation models, this paper introduces a feature disentanglement approach to improve the de-identification performance of state-of-art anonymization approaches based on voice conversion. The proposed approach achieve state-of-the-art speaker de-identification while introducting a minimal impact on the intelligibility of the signal after conversion.
Systems for analyzing the speech of signals in acoustic form have been relevant for 50 years. It is used in dialog data processing systems to recognize structure and content. Therefore, the problem of improving the quality of analogue-digital transformation is an urgent problem. The importance of the problem has increased recently due to the problem of ensuring the protection and anomination of the pores of the person's voice. For this purpose, the following has been developed: 1. methods of changing the voice (timbre, speed, level); 2. conversion of the dynamic spectrum of audio signals for face masking; 3. the scheme of the information and spectral structure of the voice (Fig. 2) on the basis of the method of adaptive self-control is substantiated; 4. a diagram of the process of decoupling the input signal and modified to assess the discrepancy has been developed;
5. A list of literature sources is provided, but there are no references to monographs on structural analysis and methods of recognition and spectral processing of speech rates.
Author Response
Reviewer full comment:
"Speech is a crucial source of personal information, and the risk of attackers using such information increases day by day. Speaker privacy protection is crucial, and various approaches have been proposed to hide the speaker’s identity. One approach is voice anonymization, which aims to safeguard speaker identity while maintaining speech content through techniques voice conversion or spectral feature alteration. The significance of voice anonymization has grown due to the necessity to protect personal information in applications such as voice assistants, authentication, and customer support. Building upon the S3PRL-VC toolkit and on pre-trained speech and speaker representation models, this paper introduces a feature disentanglement approach to improve the de-identification performance of state-of-art anonymization approaches based on voice conversion. The proposed approach achieve state-of-the-art speaker de-identification while introducting a minimal impact on the intelligibility of the signal after conversion.
Systems for analyzing the speech of signals in acoustic form have been relevant for 50 years. It is used in dialog data processing systems to recognize structure and content. Therefore, the problem of improving the quality of analogue-digital transformation is an urgent problem. The importance of the problem has increased recently due to the problem of ensuring the protection and anomination of the pores of the person's voice. For this purpose, the following has been developed: 1. methods of changing the voice (timbre, speed, level); 2. conversion of the dynamic spectrum of audio signals for face masking; 3. the scheme of the information and spectral structure of the voice (Fig. 2) on the basis of the method of adaptive self-control is substantiated; 4. a diagram of the process of decoupling the input signal and modified to assess the discrepancy has been developed;
5. A list of literature sources is provided, but there are no references to monographs on structural analysis and methods of recognition and spectral processing of speech rates"
Reply:
We thank the reviwer for the feedback on our manuscript. As far as we understand, in the last point the reviewer is asking us to expand the analysis of the state of the art. Therefore we added a citation to the following article in section 2:
Byun, K.; Moon, S.; Visser, E. Highly Controllable Diffusion-based Any-to-Any Voice Conversion Model with Frame-level 477
Prosody Feature. ArXiv 2023, abs/2309.03364
Reviewer 2 Report
Comments and Suggestions for Authors
Speaker anonymization is important in speech privacy protection. This paper presents a noval and interesting speaker anonymization method. The paper is well-written overall. Here's some suggestions on the experimental part.
First, the privacy is measured through the EER of the NVIDIA speaker recognition system in this paper. But it will be better to evaluate the privacy under different attack settings. Please refer to the experiments in [1].
Second, the utility should be evaluated using an ASR trained and tested on the anonymized speech. Please refer to the experiments in [2].
[1] Srivastava, Brij Mohan Lal, et al. "Evaluating voice conversion-based privacy protection against informed attackers." ICASSP 2020.
[2] Shamsabadi, Ali Shahin, et al. "Differentially private speaker anonymization." arXiv preprint arXiv:2202.11823 (2022).
Comments on the Quality of English LanguageMinor editing of English language required
Author Response
Reviwer's comment:
Speaker anonymization is important in speech privacy protection. This paper presents a noval and interesting speaker anonymization method. The paper is well-written overall. Here's some suggestions on the experimental part.
First, the privacy is measured through the EER of the NVIDIA speaker recognition system in this paper. But it will be better to evaluate the privacy under different attack settings. Please refer to the experiments in [1].
Second, the utility should be evaluated using an ASR trained and tested on the anonymized speech. Please refer to the experiments in [2].
[1] Srivastava, Brij Mohan Lal, et al. "Evaluating voice conversion-based privacy protection against informed attackers." ICASSP 2020.
[2] Shamsabadi, Ali Shahin, et al. "Differentially private speaker anonymization." arXiv preprint arXiv:2202.11823 (2022).
Authors' Reply:
We thank the reviewer for pointing out this limitation. Indeed it would have been better to complement our experiment with the settings suggested by the reviewer. However, it was not possible to add further experiments given the very short time available for the revision. Therefore, we propose modifying the text in Sec 4. (Experimental settings) in order to mention the settings proposed by the reviewer::
"Diverse attack scenarios are discussed in [1], showing the impact of attacker's knowledge in the achievable protection. Specifically, the strongest attack
model (“informed”) assumes that the complete knowledge of the speaker anonymization method by the adversary. However, this type of analysis is beyond the scope of the present work that focus on the simple "ignorant" setup where the attacker is not aware of the anonymization processing.
However, this type of analysis is beyond the scope of the present work.
Differently from [2], our ASR setup is based on a Kaldi-based (fixed) model trained on the original LibriSpeech clean signals; although the resulting ASR may exhibit lower performance, the decision to avaoid training on annymized speech is supported by a more equitable comparison across the analyzed converson techniques, favoring methods that exhibit lower acoustic mismatch with respect to the original audio."
Reviewer 3 Report
Comments and Suggestions for Authors
The manuscript discussed speeker anonymization and the main contribution is as follows,
1. his paper introduces a feature disentanglement approach to improve the de-identiffcation performance of state-of-art anonymization approaches based on voice conversion.
The contribution is trivial, but maybe it is practical. However, the manuscript had no details on how, why, and what to do on speeker anonymization with only simple description. It is not enough as a research paper. Therefore, I cannot recommand it to be published.
1. Page 6, “In this way, the embeddings carry less information about the speaker characteristics.” Please give more details.
2. Please give a concrete diagram on the proposed approach.
3. What is the feature disentanglement approach? How does it work? Why?
Author Response
Review's Comment
"The manuscript discussed speeker anonymization and the main contribution is as follows,
1. his paper introduces a feature disentanglement approach to improve the de-identiffcation performance of state-of-art anonymization approaches based on voice conversion.
The contribution is trivial, but maybe it is practical. However, the manuscript had no details on how, why, and what to do on speeker anonymization with only simple description. It is not enough as a research paper. Therefore, I cannot recommand it to be published."
Authors' Reply
We are deeply sorry for the fact that for the reviewer our contribution is not clear. We tried to improve our paper addressing specifically the request of the reviwer as reported below. We hope that now the manuscript is of higher quality and can be accepted for publication.
Review's Comment
Page 6, “In this way, the embeddings carry less information about the speaker characteristics.” Please give more details."
Authors' Reply
To better clarify this aspect we have modified the text as follows:
"In this way the model equipped with this additional similarity loss learns to minimize in the resulting embedding the information associated to pitch or formants and in turn to speaker characteristics, as illustrated by the results reported in Section 5."
Review's Comment
"Please give a concrete diagram on the proposed approach."
Authors' Reply
Actually we believe that the concrete block diagram of the proposed approach is shown in Figure 2, for what concerns the inference part. Figure 3 shows the training phase with the disentalgenment loss.
Review's Comment
"What is the feature disentanglement approach? How does it work? Why?"
Authors' Reply
The way feature disentanglement works in our case is described in Section 3, which shows a block diagram and reports all the mathematical formulations. We added to following text and references to better contextualise our work:
"The idea of feature disentanglement is explored in various works [61-64] and aims at encoding only specific features of the incoming speech signal (e.g. for altering prosody and expressivity); in the presented scenario we investigate the possibility of minimize the speaker information provided by popular SSL representations. Therefore we summarizes the investigated approach in the anonymization domain."
[61-64] are below:
@INPROCEEDINGS{Yang2024,
author={Yang, Zeyu and Chen, Minchuan and Li, Yanping and Hu, Wei and Wang, Shaojun and Xiao, Jing and Li, Zijian},
booktitle={ICASSP 2024 - 2024 IEEE International Conference on Acoustics, Speech and Signal Processing (ICASSP)},
title={ESVC: Combining Adaptive Style Fusion and Multi-Level Feature Disentanglement for Expressive Singing Voice Conversion},
year={2024},
volume={},
number={},
pages={12161-12165},
doi={10.1109/ICASSP48485.2024.10446284}
}
@inproceedings{Pan2024PromptCodecHN,
title={PromptCodec: High-Fidelity Neural Speech Codec using Disentangled Representation Learning based Adaptive Feature-aware Prompt Encoders},
author={Yu Pan and Lei Ma and Jianjun Zhao},
year={2024},
url={https://api.semanticscholar.org/CorpusID:268876242}
}
@article{Deng2024LearningDS,
title={Learning Disentangled Speech Representations with Contrastive Learning and Time-Invariant Retrieval},
author={Yimin Deng and Huaizhen Tang and Xulong Zhang and Ning Cheng and Jing Xiao and Jianzong Wang},
journal={ArXiv},
year={2024},
volume={abs/2401.08096},
url={https://api.semanticscholar.org/CorpusID:267028150}
}
@article{Wei2023S2CDSS,
title={S2CD: Self-heuristic Speaker Content Disentanglement for Any-to-Any Voice Conversion},
author={Pengfei Wei and Xiang Yin and Chunfeng Wang and Zhonghao Li and Xinghua Qu and Zhiqiang Xu and Zejun Ma},
journal={INTERSPEECH 2023},
year={2023},
url={https://api.semanticscholar.org/CorpusID:260916669}
}
Round 2
Reviewer 2 Report
Comments and Suggestions for Authors
The authors have answered my comments appropriately.
Author Response
We are glad that our replies were in line with the reviewer's requests.
We would like to thank once again the reviwer for the valuable comments provided during the first review round.
Reviewer 3 Report
Comments and Suggestions for Authors
The manuscript is a revised draft, which discussed speeker anonymization. The main contribution is that the manuscript introduced a feature disentanglement approach to improve the de-identiffcation performance of anonymization approaches based on voice conversion.
Suggestions:
1. Please describe which features are removed after disentangling.
2. In figure 3, WavLM is pretrained, and Lcosine is dependent on the Xe and its estimate. Therefore, Lcosine cannot be changed in the training. How does the eq.2 and 3 work?
Author Response
As in the previous review round we provide our reply to the reviewer comments below. Reviewer comments are in italics.
- Please describe which features are removed after disentangling.
Actually, since we are using neural embeddings, we do not remove any specific feature. Our process forces the embeeding extractor (WavLM) to learn representations which carry less information about the speaker characteristics so that it is more difficult to recognise the speaker identity from those embeddings.
We understand that the text may be a bit misleading, so we modified the following text in page 5:
"The idea of feature disentanglement is explored in various works [61 –64 ] and aims at encoding only specific features of the incoming speech signal (e.g. for altering prosody and expressivity); in the presented scenario we investigate the possibility of minimizing the speaker information provided by popular SSL representations. We therefore summarize the investigated approach in the anonymization domain."
by:
- replacing features with properties
- adding the following text: "Therefore the disentanglement modifies the entire hidden-state representation, learning a suitable transformation that minimizes the differences due to speaker characteristics" - In figure 3, WavLM is pretrained, and Lcosine is dependent on the Xe and its estimate. Therefore, Lcosine cannot be changed in the training. How does the eq.2 and 3 work?
WavLM is pre-trained but not frozen.Nothing is frozen in our setting.
Therefore, the Lcosine loss, fine-tuning WavLM togheter with the CTC loss, generates a transformed embedding during the disentanglement training step to uncorrelate the embeddings Xe from the speaker-related characteristics.